# Phospholipid Scramblase Activity of VDAC Dimers: New Implications for Cell Death, Autophagy and Ageing

**DOI:** 10.3390/biom14101218

**Published:** 2024-09-26

**Authors:** Patrick Rockenfeller

**Affiliations:** Chair of Biochemistry and Molecular Medicine, Center for Biomedical Education and Research (ZBAF), University of Witten/Herdecke (UW/H), Stockumer Str. 10, 58453 Witten, Germany; patrick.rockenfeller@uni-wh.de; Tel.: +49-2302-926144

**Keywords:** VDAC, porin, scramblase, apoptosis, cell death, autophagy, ageing

## Abstract

Voltage-dependent anion channels (VDACs) are important proteins of the outer mitochondrial membrane (OMM). Their beta-barrel structure allows for efficient metabolite exchange between the cytosol and mitochondria. VDACs have further been implicated in the control of regulated cell death. Historically, VDACs have been pictured as part of the mitochondrial permeability transition pore (MPTP). New concepts of regulated cell death involving VDACs include its oligomerisation to form a large pore complex in the OMM; however, alternative VDAC localisation to the plasma membrane has been suggested in the literature and will be discussed regarding its potential role during cell death. Very recently, a phospholipid scramblase activity has been attributed to VDAC dimers, which explains the manifold lipidomic changes observed in VDAC-deficient yeast strains. In this review, I highlight the recent advances regarding VDAC’s phospholipid scramblase function and discuss how this new insight sheds new light on VDAC’s implication in regulated cell death, autophagy, and ageing.

## 1. Introduction

Voltage-dependent anion channels (VDACs) are important proteins of the outer mitochondrial membrane (OMM). VDACs consist of 19 beta-sheets forming a beta-barrel structure [1]. The beta-barrel structure facilitates integration into the OMM and provides a hydrophilic channel inside the OMM, which allows for efficient metabolite exchange between the cytosol and mitochondria. VDACs are thus mainly responsible for the permeability of the OMM to small molecules. VDAC has been subject to extensive research since its first discovery in 1976 [2]. A detailed historical review of VDAC research is available and highly recommended for reading [3]. Nevertheless, the list of VDAC functions is still growing, as demonstrated by the recent finding that VDAC dimers have phospholipid scramblase activity [4]. A PubMed search for “voltage dependent anion channel” on August 22nd 2024 resulted in 3437 overall hits, which gives a good picture of the huge body of literature published on VDAC and is indicative of the broad research interest in VDAC functions, which relate to multiple human diseases such as Alzheimer’s and Parkinson’s diseases, atherosclerosis, diabetes, lupus erythematosus (SLE), colitis, and cancer [5,6].

### 1.1. Phospholipid Scramblases

All biological membranes consist of amphipathic phospholipids (PLs) forming a bilayer. The monolayers face each other through hydrophobic interactions of their inward-facing acyl chains. Sterical parameters of phospholipids are crucial for bilayer formation and function. The ideal PL to form planar bilayers is phosphatidylcholine (PC) as it comes in a near-cylindrical shape that energetically favours bilayer formation [7]. In a membrane, the two PL monolayers are coupled with each other. This means that an asymmetric distribution of PLs throughout the two leaflets triggers membrane bending. An accumulation of PLs with bulky acyl chains requiring more space and a relatively smaller headgroup bends the membrane into a convex shape, whereas cone-shaped PLs with slim hydrophobic moieties such as lyso-PLs bend towards the opposite direction. Since spontaneous PL flipping from one leaflet to the other is restricted by a considerable energy barrier, biological systems have evolved mechanisms to facilitate PL flipping to precisely regulate phospholipid exchange between monolayers [8]. Enrichment of phospholipids in one leaflet against a gradient naturally requires energy in the form of ATP as is the case for flippases and floppases, whereas phospholipid scramblases are entropy-driven enzymes which work towards an equilibration of phospholipid distribution across bilayers. This thermodynamic equilibrium is, however, only achieved if the kinetic energy barrier can be overcome. The main problem for phospholipid flipping from one leaflet to the opposite is the hydrophilic headgroup which needs to be protected from contact with the hydrophobic environment of the membrane. As for our current understanding of scramblases, these proteins enfold their enzymatic capacity mainly by two mechanisms. (1) By membrane-thinning, which means creating local areas with shorter distances and (2) by providing a hydrophilic groove reducing the energy barrier for phospholipid headgroups to transit the otherwise hydrophobic part of the bilayer. This mechanism was first suggested by Pomorski & Menon as the so-called “credit card model”, as this process has similarity to swiping a credit card through a card reader in the macroscopic world [4,8]. Scramblase activities have been identified for a number of alpha-helical proteins such as the Ca^2+^-activated TMEM16 family members [9], G-protein-coupled receptors (GPCRs) [10], the autophagy-relevant scramblase Atg9 [11,12], and bacteriorhodopsin [13]. Very recent research results describe VDAC dimers to act as lipid scramblases. VDACs are thus the first beta-barrel proteins to be attributed a phospholipid scramblase function [4].

### 1.2. Lipid Transport in Mitochondria: VDAC Dimers Facilitate Phospholipid Scrambling in the OMM

Mitochondrial membranes largely differ from other cellular membranes due to their distinguished evolutionary origin. The endosymbiont hypothesis pictures the establishment of mitochondria as cell organelles as an evolutionary process evolved from endosymbiotic α-proteobacteria, which delivers a convincing explanation of the characteristic inner mitochondrial membrane (IMM) lipid pattern, which shows striking similarities to α-proteobacterial membranes [3,14,15]. The IMM is particularly special, containing high amounts of the mitochondrial signature lipid cardiolipin (CL) together with phosphatidylethanolamine (PE) and phosphatidylglycerol (PG). CL and PE, which together make up more than 50% of the IMM’s lipid content, are both non-bilayer lipids providing a negative curvature to membranes [7,16]. This unique feature largely defines the properties of the IMM, which is characterised by high numbers of invaginations (cristae), thus creating a huge surface to accommodate the large protein complexes of the respiratory chain and ATP synthesis [17]. Importantly, the enzymatic endowment of mitochondria is limited to the production of PE, PG, and CL, and relies on the supply of the phospholipid precursors phosphatidic acid (PA) and phosphatidylserine (PS) from other cell organelles. The ER is the major source of cellular phospholipid synthesis and is, therefore, the main PA and PS supplier for mitochondrial usage. The molecular mechanisms facilitating PA transport from the OMM to the IMM have been resolved very elegantly by the Langer laboratory and are no longer enigmatic [18,19,20]. In yeast, PA transport is facilitated by a heterodimeric complex of the lipid transfer protein Ups1 and Mdm35 [18,21]. This complex is conserved in humans, where it is called PRELID1/TRIAP1 [22]. PS transport in yeast relies on Ups2 and its association with Mdm35 [19,20]. The conserved human orthologues of Ups2 are PRELID3A and PRELID3B, which, in association with TRIAP1, facilitate PS transport in human mitochondria [19].

Even though the molecular details of PS and PA transport from the OMM to the IMM were well understood, one important question remained. It was the question of how PA, PS, PE, or other phospholipid molecules could flip between the outer and inner phospholipid leaflets of the OMM. A number of studies had already observed that leaflet flipping in the OMM occurred independently of ATP, which suggested the existence of an OMM-residing phospholipid scramblase [23,24,25,26]. With the recent article published by Jahn et al. from the Menon laboratory, such phospholipid scramblase activity has been attributed to homodimers of VDAC1 and 2 in humans and homodimers of their budding yeast orthologues, Por1 and Por2 [4]. Scramblase activity was measured in large unilamellar vesicles harbouring reconstituted VDAC proteins using two different scramblase assays. One scramblase assay made use of phospholipase C activity, which can only access phospholipids in the external liposome leaflet. By quantifying radiolabelled phosphatidylinositol degradation, any degradation exceeding 50% was attributed to phospholipid activity resulting from reconstituted VDAC proteins. The second scramblase assay involved BSA-mediated fluorescence quenching of NBD-PC following a similar rationale; that is, only PC from the external liposomal leaflet is subject to BSA-mediated extraction, thus leading to fluorescence quenching. Any additional quenching can be attributed to PL scramblase activity [4]. Por1- and Por2-dependent scramblase activity was further assayed in purified yeast mitochondria, establishing a four-state kinetic model [4]. The scramblase activity of VDAC dimers is thus evolutionarily conserved from yeast to humans, indicating that this activity is an important function for cellular lipid homeostasis and cell health. In fact, Por1 and Por2 were identified as the predominant lipid scramblases in the OMM, accounting for at least 90% of mitochondrial lipid import through the OMM in yeast [4]. Dimer formation was verified by crosslinking and suggested a parallel association of two VDAC1 monomers through the interaction of beta strands 1, 2, 18, and 19. Mechanistically, these VDAC1 dimers are thought to facilitate phospholipid flipping by membrane thinning and by providing a hydrophilic groove through hydrophilic amino acid residues at the dimer interface. These hydrophilic amino acid residues at the dimer interface include T77, S43, T33, S35, Y247, and Q249. VDAC1 dimers where all of these polar residues had been replaced with valine showed poor scramblase activity in molecular dynamics simulations, suggesting that they are indeed key to the phospholipid scramblase activity of VDAC1 dimers [4].

The evidence for VDAC/Por-phospholipid scramblase function is thus very convincing and delivers very suitable explanations for the lipidomic changes observed in earlier studies using *por1*∆, *por2*∆, and *por1*∆ *por2*∆ double deletion strains in yeast, which include reduction in CL and PE [27,28,29].

### 1.3. VDAC-Related Mitochondrial Lipid Changes Affect Autophagy

The mitochondrial lipid composition not only affects the activity of mitochondrial membrane protein complexes, which largely rely on interactions with specific local lipid environments [30], the changes in PE production also affect the efficiency of autophagy [28,31,32], which represents a pro-survival pathway during cellular ageing [33,34,35]. Autophagy is characterised by the generation of an isolation membrane (IM), which extends in size to engulf cellular materials in a double membrane structure called the autophagosome [36,37,38]. Autophagosome fusion with the lysosome (vacuole in yeast) results in the degradation of autophagosomal cargo. Autophagy can thus be envisioned as a cellular quality control program protecting against cellular damage by clearing adverse compounds, with beneficial effects on cellular ageing [34,35]. Autophagic membranes are generated in a complex manner involving multiple organelles, including the ER [39], mitochondria [40], Golgi [41,42], plasma membrane [43,44], endosomes [45,46], and lipid droplets [47,48]. These organelle-derived lipids are predominantly delivered via Atg9-containing vesicles and can be regarded as seeds for autophagosomal membrane formation [49].

Autophagosomal membrane generation not only includes lipid transfer from pre-existing organelles but the extension of autophagosomal membranes further requires de novo phospholipid synthesis, which is facilitated through close interaction with the ER at ER–autophagosome contact sites [50]. Lipid exchange at the ER–autophagosome contact site occurs through a lipid channel formed by the Atg2-WIPI/Atg18 complex [51,52,53,54,55]. Activated fatty acids are channelled into the ER, where they feed into phospholipid generation, which can be used for autophagosomal membrane expansion [50]. Atg9 functions as a phospholipid scramblase that equilibrates phospholipid distribution across membrane leaflets [11]. The availability of fatty acids [56,57,58] and phosphatidylethanolamine (PE) [32] have further been shown to be limiting factors for autophagy. It thus can be constituted that mitochondria in general [59], and phosphatidylserine decarboxylase 1 (Psd1) [32] and VDAC/Por [28] in particular, affect cellular autophagy regulation. The direct influence of Psd1 on PE levels can be easily explained by the need for PE in autophagosomal membrane biogenesis since PE together with PC and PI make up the majority of the phospholipid content of autophagosomes [50]. However, the rationale for VDAC/Por’s mechanistic role in autophagy remained somehow blurry. With the evidence of VDAC/Por dimers acting as phospholipid scramblases [4], a satisfactory explanation is now available: we hypothesise that the phospholipid scramblase activity of VDAC/Por dimers is responsible for the influence on autophagy and thus the predominant reason why deletion of VDAC/Porin reduces the autophagic capacity (Figure 1). We reason that (1) PS accesses the OMM leaflet facing the intermembrane space through VDAC dimer scramblase activity. (2) Psd1 catalyses the reaction from PS to PE by decarboxylation. (3) PE is transported to the ER via ER–mitochondria contact sites such as MAMs or ERMES. (4) PE is further channelled into autophagosomal IMs through the lipid channel formed by Atg2-WIPI/Atg18 (Figure 1). This model thus offers a suitable explanation for why VDAC deficiency results in reduced autophagic capacity. The reduced autophagic activity in *por1*-deleted yeast strains can be partially attenuated by Psd1 overexpression [28]. Since Psd1 can be dually targeted to mitochondria and the ER [60], it remains to be investigated to what extent these different Psd1 pools (ER or mitochondria) affect autophagy. Importantly, the relevance of these different Psd1 pools might also depend on the type of autophagy and the type of induction (e.g., nitrogen starvation, rapamycin treatment, spermidine administration, glucose/energy deprivation, fatty acid administration, etc.), which involve different phenotypes and different sets of autophagic machinery [36,38,57,61,62,63].

Very recently, a new role for VDAC in piecemeal removal of the IMM has been identified [64]. The data suggests that this microautophagy-like partial degradation of single cristae IMM material can be facilitated by extrusion through an oligomeric VDAC channel. The IMM material would turn inside out through the oligomeric VDAC pore, be cleaved by the endosomal sorting complex required for transport (ESCRT) machinery, and be degraded via fusion with lysosomes [64,65]. The newly identified mechanism shows how elegant and economic cellular quality control mechanisms have developed.

### 1.4. VDAC and Its Role in Cell Death

Besides their important functions in energy metabolism, mitochondria are further implicated in the coordination of regulated cell death processes [66]. Importantly, the intrinsic pathway of apoptosis is largely connected to the release of mitochondrial factors [67] such as apoptosis-inducing factor 1 (aif1) [68], mitochondrial DNA [69,70], cytochrome c [71], second mitochondria-derived activator of caspase (SMAC) [72], and direct inhibitor of apoptosis-binding protein with low pI (Diablo) [5,73,74,75]. The release of mitochondrial effectors is achieved through mitochondrial outer membrane permeabilization (MOMP). MOMP is facilitated by the proapoptotic proteins BAX (apoptosis regulator X) and BAK (BCL2 antagonist/killer), which are absolutely required for MOMP during intrinsic apoptosis [76]. BOK (Bcl-2 ovarian killer) [77] and BID [78] also have the capacity to form apoptotic pores within the OMM. Super-resolution microscopy revealed the formation of rings and arcs by BAX oligomerisation, which is key to mitochondrial pore formation [79].

Historically, VDACs have been discussed as potential candidates, together with adenine nucleotide transporters (ANTs) and cyclophilin D, forming the mitochondrial permeability transition pore (mPTP) [80]. Even though the precise molecular architecture of the mPTP is still unsolved, it has become clearer that the mPTP is predominantly formed through dimerisation of the F_0_F_1_-ATP synthase and that it does not require VDACs or ANTs to function [81,82]. It has been hypothesised that VDAC1 has the capacity to form higher-order oligomeric pore structures in the OMM during apoptosis, either by itself or in combination with BAX [5,73,74]. If such large mitochondrial pores formed by VDAC1 with or without the involvement of BAX truly existed in vivo, they could permit mitochondrial release of proapoptotic factors such as those mentioned above.

### 1.5. Novel Hypotheses Relating to VDAC Function as a Phospholipid Scramblase

#### 1.5.1. Hypothesis 1

Even though VDACs are predominantly described as channels of the OMM, a few studies have identified VDACs in other cellular compartments, including the ER/SR, Golgi and the plasma membrane [83]. These alternative VDAC localisations are described particularly under conditions of stress or disease. Regarding the novel finding of VDAC dimers acting as phospholipid scramblases, the plasma membrane localisation under apoptotic conditions offers an appealing hypothesis: during apoptosis, the lipid asymmetry of the plasma membrane regarding PS content is lost [66]. PS externalisation to the external leaflet is an important regulatory feature of apoptotic cells attracting macrophages [66,84]. Some studies suggest scramblase activities of PLSCR, Xkr, and TMEM families in combination with a loss of flippase/floppase activities as mediators of PS externalisation during apoptosis [66,85,86,87,88,89,90]. Combining the two concepts/hypotheses of (1) VDAC dimers acting as lipid scramblases and (2) alternative descriptions of VDAC cellular localisation to the plasma membrane under conditions of apoptosis, it should be considered that VDAC dimers could act as phospholipid scramblases mediating PS externalisation at the plasma membrane during apoptosis (Figure 2). Of note, this hypothesis has highly speculative character as it creates additional questions/problems, for example, why cells harbouring VDACs in their plasma membranes are not affected by its function of ion conductance.

#### 1.5.2. Hypothesis 2

An alternative hypothesis of VDAC scramblase function during apoptosis suffices a purely mitochondrial localisation. The PRELID/TRIAB complex (as described above) has been characterised as having antiapoptotic properties since its abrogation led to increased susceptibility to apoptotic stimuli [22]. This antiapoptotic trait is thought to be linked to its PA transport activity, which provides PA as a precursor for mitochondrial CL synthesis. PA conversion to CL involves four enzymes, namely, TAM41 mitochondrial translocator assembly and maintenance homolog (TAMM41), phosphatidylglycerophosphate synthase (PGS1), protein tyrosine phosphatase mitochondrial 1 (PTPMT1), and cardiolipin synthase (CLS1) [20]. Choline-diphosphate-diacylglycerol (CDP-DAG), phosphatidylglycerol-phosphate (PGP), and PG are the lipid intermediates of these reactions. CL further undergoes acyl chain remodelling, which is catalysed by Taz1 enzyme activity [91]. The mitochondrial nucleoside diphosphate kinase NDPK-D and mitochondrial creatine kinase (MtCK) have the capacity to transfer CL from the IMM to the OMM as part of a stress response [20,92,93,94]. An overall reduction in mitochondrial CL has been described to render cells susceptible to apoptotic stimuli because CL is needed for cristae-structure and protein-supercomplex stabilisation in the IMM [30]. We thus hypothesise that VDAC can regulate mitochondrial apoptosis and mitophagy through its phospholipid scramblase activity in the OMM. One possible explanation is that the assembly of scramblase-active VDAC dimers would allow for PA and PS access to mitochondria, whereas monomeric VDAC states could limit the access of these important lipid precursors in mitochondria and thereby stimulate apoptosis or increase susceptibility to apoptotic stimuli. The control of VDAC oligomerisation status would thus be a regulatory handle for cell health and quality control. Whether OMM VDAC scramblase activity could also succumb to cell death regulation via flipping of CL or oxidised CL species to the cytosol-facing leaflet of the OMM is unknown. This feature (CL flipping) has been attributed to the mitochondrial phospholipid scramblase 3 (PLS3). Importantly, CL exposure at the OMM has been described to occur upon stress during mitochondrial apoptosis and as a hallmark of mitophagy [30,95] (Figure 3). Activation of caspase 8 and BAX are downstream apoptotic events [20,94,96], whereas CL recognition by the autophagy-related protein microtubule-associated protein 1 light chain (LC3) is a feature of mitophagy [97].

#### 1.5.3. Hypothesis 3: VDAC Scramblase Activity at the Plasma Membrane in Alzheimer’s Disease

During Alzheimer’s disease (AD), Amyloid β (Aβ) is produced from the amyloid precursor protein (APP). AD pathogenesis includes Aβ release to the extracellular space, where its aggregation causes the formation of senile plaques. VDAC1 has been shown to associate with APP and Aβ in lipid rafts of neurons [98,99,100]. During Aβ processing, APP gets palmitoylated and translocates to lipid rafts in the plasma membrane [101]. It might be worth investigating whether VDAC dimers could assist with APP insertion into the plasma membrane either by providing a suitable lipid environment or by inserting the lipophilic APP into the plasma membrane itself. Molecular dynamics simulations might deliver first insights into whether such hypotheses turn out to be robust.

## 2. Discussion and Open Questions

Recent research advances hypothesise that the dimerisation/oligomerisation status of VDAC is ultimately linked to specific functions [4,5,64,73]. The obvious big question that arises is of course “What determines the VDAC dimerisation/oligomerisation status?” Is it VDAC phosphorylation? For example, Bobba et al. suggested that AMPK activation during apoptosis of cerebellar granule cells influences VDAC1 phosphorylation and activity [102]. To answer this important question, a robust assay to detect VDAC homomerisation is needed. One such assay is the homodimeric protein complementation assay (hdPCA), which is based on dihydrofolate reductase (DHFR) complementation, as established in the yeast model system by Stephen Michnick’s laboratory [103,104]. A multitude of conditions could be easily screened in this system to identify such conditions, which trigger Por1 dimerisation. As part of such a screen, the second open question of whether VDAC/Por dimerisation affects cell death regulation could be tackled as well.

Another interesting question is whether VDAC/Por dimerisation can be pictured as a dynamic process, meaning that switches between monomeric, dimeric, and oligomeric statuses fluctuate or oscillate and are responsive to diverse cellular conditions. How does VDAC/Por dimerisation/oligomerisation develop throughout ageing? Is there a change that can be causally linked to ageing? Is lipid availability a control parameter of VDAC oligomerisation status? Is it only homodimers, or are heterodimers, e.g., between the VDAC isoforms VDAC1, VDAC2, and VDAC3 (or Por1/Por2, Por1/Mdm10, and Por/Mdm10 in yeast) also involved in lipid scrambling?

All these open questions are highly interesting, and answering them could have a huge impact on our understanding of human biology and ageing; this should give investigations in this field high priority.

## 3. Conclusions

To conclude, we can say that the central mitochondrial player, VDAC, and its involvement in cell death, autophagy, and ageing is of outstanding interest to the research community. In particular, its role in lipid homeostasis and membrane contact site formation are recent research topics of high priority with major relevance to understanding human health and disease.

## Figures and Tables

**Figure 1 biomolecules-14-01218-f001:**
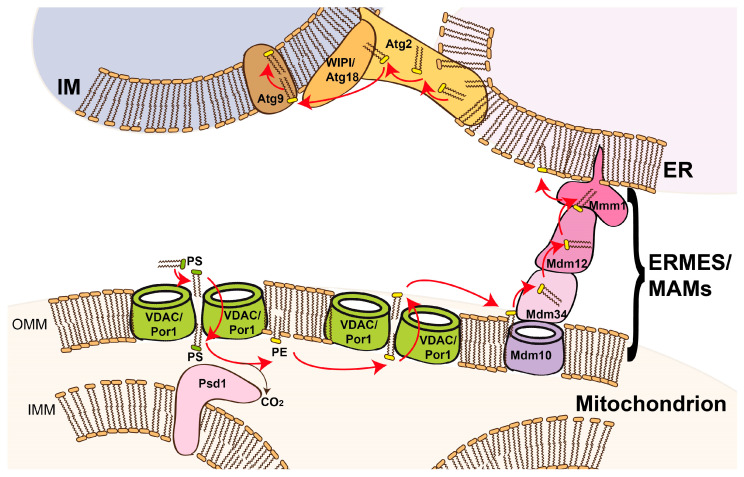
Model of VDAC/Por1 scramblase function during autophagy. Dimeric VDAC/Por1 effectively flips PS from the cytosol-facing leaflet to the IMS-facing leaflet of the OMM through its scramblase function, as successfully flipped PS is withdrawn from the equilibrium by Psd1-mediated decarboxylation acting from the IMM in trans. PE can be re-externalised to the cytosol-facing leaflet by VDAC/Por1 scramblase activity and channelled into the ER via ER–mitochondria contact sites such as ERMES or MIA in yeast or MAMs in mammalian cells. PE situated in the outward-facing ER leaflet can be transported to autophagosomal IMs via ER exit sites (ERES) via Atg2/Atg18, which functions as a lipid channel. Atg9 scrambles phospholipids across IM leaflets.

**Figure 2 biomolecules-14-01218-f002:**
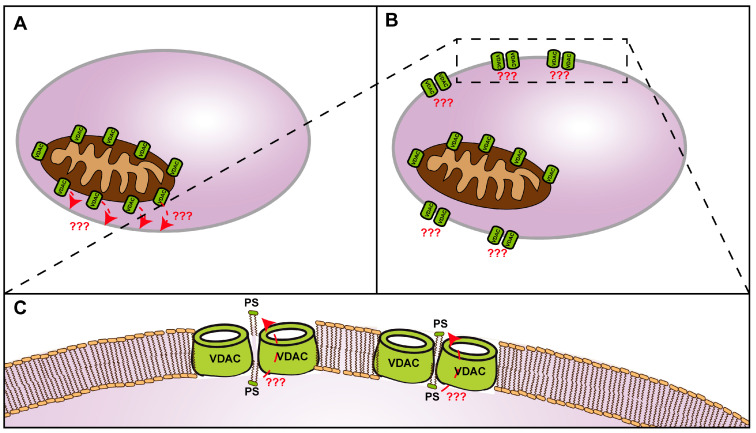
Model of potential VDAC scramblase activity in the plasma membrane during apoptosis. VDAC/Por1 is overexpressed under diverse pathological conditions associated with apoptosis. (**A**) VDAC/Por1 overexpression might trigger its implementation and dimerisation in the plasma membrane (**B**) where it could externalise PS through its phospholipid scramblase activity (magnification in (**C**)). Dashed arrows and question marks indicate hypothetical character of events.

**Figure 3 biomolecules-14-01218-f003:**
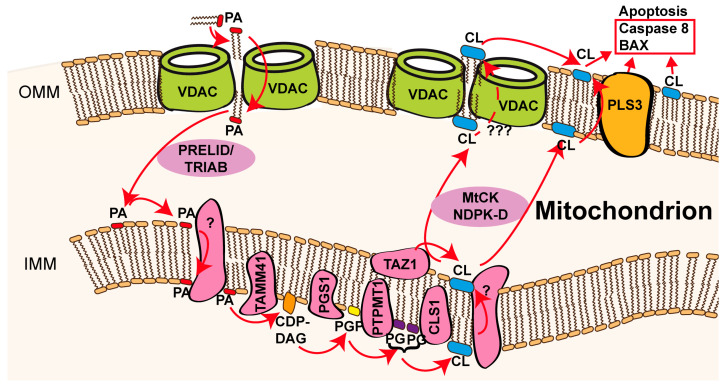
Model of potential involvement of VDAC dimer scramblase activity in mitochondrial apoptosis/mitophagy. VDAC/Por1 dimers scramble phospholipids in the OMM, which leads to the de facto import of PA from the cytosolic-facing to the intermembrane-space-facing membrane leaflet. PRELID/TRIAB transports PA from the OMM to the IMM, reducing its concentration at the IMM-facing leaflet of the OMM, which drives directed scramblase activity towards import. PA accesses the matrix-facing leaflet of the IMM via an unknown lipid scramblase or flippase. PA is converted to CL in four steps that involve choline-diphosphate-diacylglycerol (CDP-DAG), phosphatidylglycerol-phosphate (PGP), and phosphatidylglycerol (PG), and is catalysed by consecutive activity of TAM41 mitochondrial translocator assembly and maintenance homolog (TAMM41), phosphatidylglycerophosphate synthase (PGS1), protein tyrosine phosphatase mitochondrial 1 (PTPMT1), and cardiolipin synthase (CLS1). After scrambling/flipping activity (from unknown protein function), CL undergoes acyl chain remodelling by TAZ1 activity and is then transported to the intermembrane-space-facing membrane leaflet of the OMM, which depends on mitochondrial nucleoside diphosphate kinase (NDPK-D) and mitochondrial creatine kinase (MtCK). Phospholipid scramblase 3 (PLS3) enables the flipping of CL to the outer OMM leaflet, a reaction that might alternatively also be executed by VDAC dimers, as indicated by the dashed arrow and question marks.

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
