# Peer review of "Phospholipid Scramblase Activity of VDAC Dimers: New Implications for Cell Death, Autophagy and Ageing"

_biomolecules, 2024, doi:10.3390/biom14101218_

Round 1
Reviewer 1 Report
Comments and Suggestions for Authors
RE: “Phospholipid scramblase activity of VDAC dimers – new implications for cell death, autophagy and ageing” by Patrick Rockenfeller
TO THE AUTHOR
This is an informative review on the new concepts VDAC scramblase activity and other VDAC functions assigned to its oligomerized states. The author gives a comprehensive picture of modern hypotheses of VDAC´s involvement in regulated cell death and autophagy with the main stress on VDAC’s newly found ability to facilitate exchange of lipid molecules between opposing membrane leaflets.
My major point of criticism is that some of the concepts, which author introduces as proven and well-accepted by using “it was shown” or similar wording for their description, are not such. Most of all, this concerns the pore-forming VDAC oligomerization that is based merely on the cartoons repetitively reproduced by the same authors in different publications. To a smaller degree, it also concerns the existence of VDAC pores in plasma membranes. The proper wording would be “hypothesized”, “assumed”, “suggested”, etc. If not corrected, the review can be misleading and thus doing harm to the field.
Other, mostly smaller points:
Line 10, Abstract: “Their sophisticated beta barrel structure” – beta-barrel structures are usually considered to be relatively simple compared to the structures of multi-domain proteins, not "sophisticated".
Line 26: “VDACs consist of 15 beta-sheets” – it is usually 19 beta-strands, not 15 beta-sheets.
Lines 50-51: “…cone-shaped PL with slim hydrophobic moyeties” – do you mean "hydrophobic moieties"?
Lines 157-159: “Since Psd1 can be dually targeted to mitochondria and the ER [60] it remains to be investigated, to which extent…” – move the comma to after “ER [60]”.
Lines 198-199: “VDAC1 has the capacity to form higher order oligomeric pore structures in the OMM” – as I mentioned above the only ‘proof’ of this statement is the cartoons produced and reproduced by basically the same group of authors. It is still a proclamation, not an established fact. I suggest introducing this statement by “It was hypothesized that…”
Lines 207-208: “…numerous studies have identified VDACs in other cellular compartments…” – not numerous, a few. Besides, if it were true, one should see a surge of calcium (and other ions) into cells long before PS externalizes, because VDAC, even in its "closed" state, is quite conductive. Such a surge was not observed.
Figure 3 and discussion around it: Why bring VDAC in CL flip when PLS3 is involved?
Line 291: “Recent research advances have shown that the dimerisation/ oligomerisation status of VDAC…” – Better "hypothesized", not "have shown" (see above).
Author Response
This is an informative review on the new concepts VDAC scramblase activity and other VDAC functions assigned to its oligomerized states. The author gives a comprehensive picture of modern hypotheses of VDAC´s involvement in regulated cell death and autophagy with the main stress on VDAC’s newly found ability to facilitate exchange of lipid molecules between opposing membrane leaflets.
My major point of criticism is that some of the concepts, which author introduces as proven and well-accepted by using “it was shown” or similar wording for their description, are not such. Most of all, this concerns the pore-forming VDAC oligomerization that is based merely on the cartoons repetitively reproduced by the same authors in different publications. To a smaller degree, it also concerns the existence of VDAC pores in plasma membranes. The proper wording would be “hypothesized”, “assumed”, “suggested”, etc. If not corrected, the review can be misleading and thus doing harm to the field.
I am particularly thankful for this comment and adapted the manuscript accordingly.
Other, mostly smaller points:
Line 10, Abstract: “Their sophisticated beta barrel structure” – beta-barrel structures are usually considered to be relatively simple compared to the structures of multi-domain proteins, not "sophisticated".
The word “sophisticated“ has been removed.
Line 26: “VDACs consist of 15 beta-sheets” – it is usually 19 beta-strands, not 15 beta-sheets.
That was a slip. Of course it is 19 beta-strands, which has been corrected.
Lines 50-51: “…cone-shaped PL with slim hydrophobic moyeties” – do you mean "hydrophobic moieties"?
Yes, it´s “moieties”, which has been corrected now.
Lines 157-159: “Since Psd1 can be dually targeted to mitochondria and the ER [60] it remains to be investigated, to which extent…” – move the comma to after “ER [60]”.
Is corrected.
Lines 198-199: “VDAC1 has the capacity to form higher order oligomeric pore structures in the OMM” – as I mentioned above the only ‘proof’ of this statement is the cartoons produced and reproduced by basically the same group of authors. It is still a proclamation, not an established fact. I suggest introducing this statement by “It was hypothesized that…”
I have adapted the statement as suggested.
Lines 207-208: “…numerous studies have identified VDACs in other cellular compartments…” – not numerous, a few. Besides, if it were true, one should see a surge of calcium (and other ions) into cells long before PS externalizes, because VDAC, even in its "closed" state, is quite conductive. Such a surge was not observed.
The paragraph has been adapted.
Figure 3 and discussion around it: Why bring VDAC in CL flip when PLS3 is involved?
CL flipping in the OMM has been described to occur through PLS3 in the literature, which does not exclude the potential use of other scramblases, which is why I mention that VDAC dimers might have additional activity here. I visualised this potential involvement with a dotted line and included question marks to indicate its hypothetical nature.
Line 291: “Recent research advances have shown that the dimerisation/ oligomerisation status of VDAC…” – Better "hypothesized", not "have shown" (see above).
This has been changed accordingly.
Reviewer 2 Report
Comments and Suggestions for Authors
In this review, the author updated the recent findings on VDAC especially on its phospholipid scramblase activity. Overall, the history of VDAC research was well summarized. The following points are recommended for consideration.
1) For better understanding of the authors, it might be better to include more detailed description of the paper that showed the phospholipid scramblase activity of VDAC. For example, the main experimental setting that led to this discovery and the actual molecular mechanisms by which VDAC dimer exhibits the activity (e.g. Are there any specific amino acid residues for this activity? Are these conserved in Yeast VDAC? etc.) .
2) Each figure contains significant amount of author’s hypothesis that is not experimentally verified. Therefore, these points should be distinguished from others to avoid misunderstanding of the readers. For example, at least, it is better to add question marks to these points.
Author Response
In this review, the author updated the recent findings on VDAC especially on its phospholipid scramblase activity. Overall, the history of VDAC research was well summarized. The following points are recommended for consideration.
1) For better understanding of the authors, it might be better to include more detailed description of the paper that showed the phospholipid scramblase activity of VDAC. For example, the main experimental setting that led to this discovery and the actual molecular mechanisms by which VDAC dimer exhibits the activity (e.g. Are there any specific amino acid residues for this activity? Are these conserved in Yeast VDAC? etc.) .
I extended paragraph 2 accordingly to further delineate the molecular details of PL scrambling and the assays which were used to identify this activity.
2) Each figure contains significant amount of author’s hypothesis that is not experimentally verified. Therefore, these points should be distinguished from others to avoid misunderstanding of the readers. For example, at least, it is better to add question marks to these points.
I have added question marks and dotted lines to the relevant steps with hypothetical character which should now allow to clearly differentiate between prooven concepts and hypotheses.
Round 2
Reviewer 1 Report
Comments and Suggestions for Authors
The author took good care of my criticism.